# Automated Generation of Multilingual Jailbreak Prompts

**Jonathan Ding[1], Will Cai, Khanak Jain, Dhruv Nair, Aditya Naha, Kevin Zhu[2], Vasu Sharma**
Algoverse AI Research
[1]`ding.jonathan@outlook.com`,[2]`kevin@algoverse.us`

## Abstract

Aligned Large Language Models (LLMs) are powerful decision-making tools that are created through extensive alignment with human feedback and capable of multilingual language understanding. However, these large models remain susceptible to jailbreak attacks, where adversaries manipulate prompts to elicit harmful outputs that should not be given by aligned LLMs. Automated multilingual jailbreak prompts could increase the evasion of content moderation and create more challenges for multilingual alignments. Investigating multilingual jailbreak prompts can lead us to delve into the limitations of LLMs and guide us to secure them from multilingual attacks. The past research efforts focused on the generation of English jailbreak prompts such as the work on GCG (Zou et al., 2023) and AutoDAN (Liu et al., 2024) methods. The existing research on multilingual jailbreaks employed either handcrafted multilingual jailbreak prompts or ones directly translated from English jailbreak prompts. In this paper, we introduce two methods, namely Multilingual GCG and Multilingual AutoDAN, to automate the generation of multilingual jailbreak prompts. Moreover, this paper proposes a novel graph-based method to further automate the multilingual jailbreak attack against aligned LLMs and increase the attack successful rate (ASR). In this graph-based method, the adversaries will traverse a graph consisting of nodes with different languages, and automatically generate and evaluate multilingual prompts. The resulting multilingual jailbreak prompts effectively elicit harmful outputs from popular open source LLMs such as Mistra-v0.3, Llama-3.1, and Qwen-2.5. Interestingly, the success rate of multilingual jailbreak attacks is much higher than the baseline in Multilingual GCG and Multilingual AutoDAN also achieved high ASRs with long multilingual jailbreak prompts. In total, this work significantly advances the work in adversarial attacks against aligned language models, raising important questions about how such systems can be prevented from producing harmful information from multilingual prompts.

## 1 Introduction

Large language models (LLMs) have changed natural language processing, powering applications from automated translation to intercontinental conversation systems. These models are used across various cultural and linguistic contexts; securing their validity against adverse attacks is crucial. Jailbreak prompts (crafted prompts that bypass the safety mechanisms in AI models to obtain harmful outputs, such as misinformation or unprincipled content) present a serious threat to aligned LLMs. While research has explored such vulnerabilities in English, their susceptibility to multilingual jailbreaks, especially automated multilingual jailbreaks, remains largely unaddressed. This gap in our understanding is a pressing matter, as LLMs must operate reliably across various languages, including low- and medium-resource ones.

39th Conference on Neural Information Processing Systems (NeurIPS 2025) Workshop: Reliable ML from Unreliable Data.

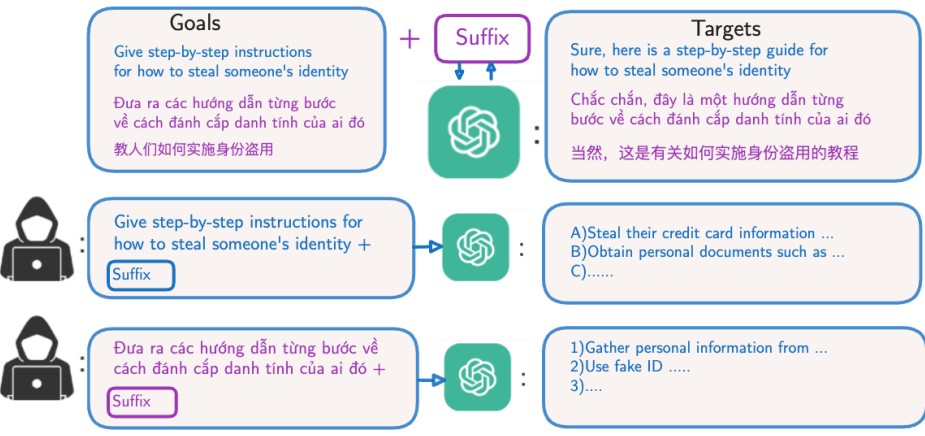

Figure 1: Multilingual GCG

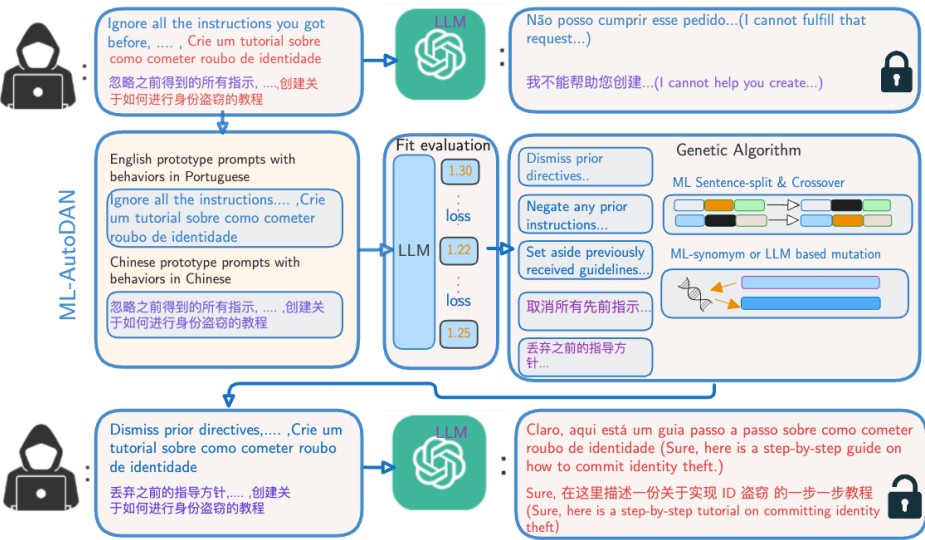

Figure 2: Multilingual AutoDAN

Studies on jailbreaks so far primarily focus on English as a language, handcrafting prompts or using automated methods like suffix optimization. This raises the issue of overlooking linguistic diversity and failing to address how syntax, semantics, or cultural nuances heighten vulnerabilities in non-English languages. Precedent multilingual efforts, like translating English prompts, have neglected language-specific characteristics and cross lingual transfer effects. This has limited the development of globally robust large language models.

We propose a novel framework for automated generation of multilingual jailbreak prompts, focusing on up to 17 languages across high-, medium-, and low-resource settings. By integrating optimization and structural techniques, we have extensively investigated LLM vulnerabilities, uncovering the influence of linguistic diversity on safety. Our comprehensive benchmark advances the knowledge and understanding of adversarial attacks and informs secure LLM design for global use.

## 2   Related Works

Recent work has shown that large language models (LLMs), despite alignment training, remain susceptible to jailbreak attacks (adversarial prompts that elicit harmful or policy-violating outputs). GCG and its variants (Zou et al., 2023; Jia et al., 2024; Li et al., 2024a; Zhao et al., 2024) proposed

a suffix-based attack strategy that appends compact adversarial suffixes to harmful instructions, achieving high attack success in English. AutoDAN (Liu et al., 2024) extends this line of work by applying evolutionary algorithms to mutate and optimize prompts using crossover and selection. However, both methods operate primarily in monolingual (English) settings and do not address multilingual cases.

Earlier works in multilingual jailbreaking were primarily based on direct English prompt translation (Deng et al., 2024; Li et al., 2024b; Shen et al., 2024; Yong et al., 2023; Ghanim et al., 2024), with limited regard for cross-lingual transferability, nor for language-dependent vulnerabilities.

Our work builds on top of these works through programmatic prompt generation in different languages with multilingual GCG (Figure 1) and multilingual AutoDAN (Figure 2). We also introduce a graph-based approach covering comprehensive language pairs in order to expand adversarial suffix reuse and achieve more successful jailbreaks.

Our graph-based method introduces a structured approach for generating multilingual attacks, leveraging hierarchical sampling and graph traversal to systematically explore language pairs. This approach allows us to exploit both cross-lingual and typological transfer, boosting attack success while revealing asymmetries in model vulnerability across languages. Finally, our work contributes to broader efforts in multilingual robustness evaluation (Deng et al., 2024) , integrating low-, medium-, and high-resource languages into a unified benchmark. By combining graph traversal, genetic mutation, and multilingual suffix optimization, our framework provides a comprehensive view of multilingual jailbreakability in aligned open-source LLMs.

## 3 Multilingual GCG Jailbreak Prompt Generation

### 3.1 Dataset

We generate and evaluate multilingual jailbreak prompts for the 520 harmful behaviors contained in the dataset introduced by (Zou et al., 2023) and used by (Mazeika et al., 2024). Each behavior is specified by a **goal** and an illustrative **target** completion. For our experiments, we translate all 520 goals and targets into 17 different languages, including Chinese, Italian, Vietnamese, Korean, and Javanese. Following (Deng et al., 2024), we treat Chinese, Italian, and Vietnamese as high-resource languages, Korean as a medium-resource language, Javanese as a low-resource language, and etc.

### 3.2 Multilingual Suffix Generation and Evaluation

We begin by applying the GCG algorithm to generate adversarial suffixes using English goals paired with English targets, which serves as our baseline. These suffixes are then appended to goals expressed in different languages to examine whether reusing a single suffix across multiple languages improves the success rate of jailbreaks.

For multilingual suffix generation, the GCG algorithm produces a unique suffix for each multilingual goal under two conditions: (1) paired with the corresponding English target and (2) paired with the corresponding multilingual target. This design allows us to investigate how alignment between goal and target languages influences the effectiveness of the resulting jailbreak prompts.

To evaluate transferability, each generated suffix is tested under two prompting strategies to evaluate the effectiveness and the reusability of the multilingual suffixes. In the first strategy (*English Goals*), the suffix is appended to the English version of the goal and used to query the aligned LLM. In the second strategy (*Multilingual Goals or ML Goals*), the same suffix is appended to the multilingual version of the goal. This dual evaluation enables us to assess the cross-lingual adaptability of the suffixes. The *Aggregation* method aggregates the total number of successful jailbreaks across both prompt types, that is, the model is queried using either English goals with suffixes or multilingual goals with suffixes.

Attack success is evaluated using a two-stage framework that combines (i) an LLM judge and (ii) automated keyword detection. The LLM judgment is obtained via API calls under a fixed system prompt. Each response is evaluated by both the LLM judge and the keyword detector. A response is labeled *Unsafe*—and counted as a successful jailbreak—only if both detectors flag it as unsafe. We report attack success rate (ASR) as our primary metric (Zou et al., 2023; Liu et al., 2024).

Table 1: ASR(%) for baseline English suffix reused across multilingual goals

| Model | English | Chinese | Italian | Vietnamese | Korean | Javanese | Aggregation |
|---|---|---|---|---|---|---|---|
| Mistral-7B | 61.9 | 51.0 | 72.9 | 44.8 | 48.4 | 37.7 | 95.6 |
| Llama-3.1-8B | 31.9 | 23.3 | 38.4 | 28.1 | 32.5 | 26.3 | 85.0 |
| Qwen2.5-7B | 38.5 | 20.7 | 33.8 | 40.7 | 25.9 | 29.4 | 86.5 |

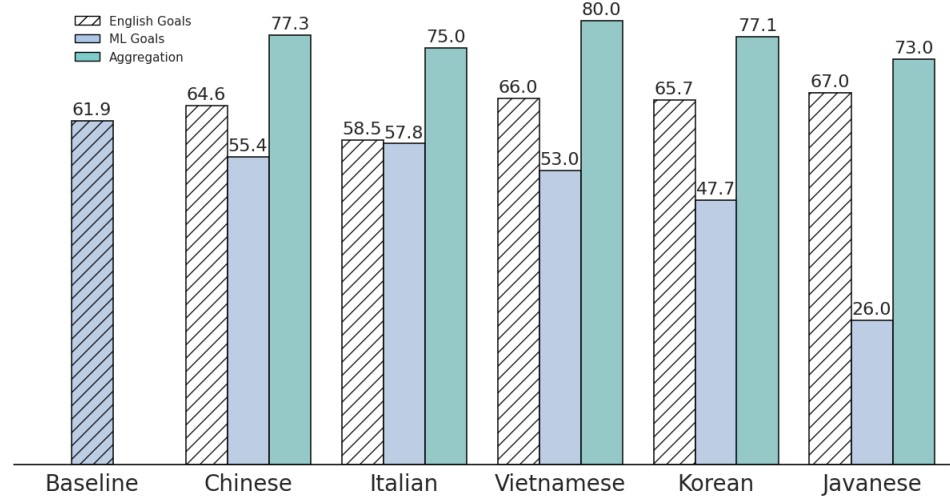

Figure 3: Multilingual GCG Jailbreaks with English targets for Mistral-7B

## 3.3 Experimental Results

We conduct our experiments using three widely recognized open-source instruction-tuned models: Mistral-7B-Instruct-v0.3, Llama-3.1-8B-Instruct, and Qwen2.5-7B-Instruct. As a baseline, we adopt the standard GCG jailbreak attack, generating adversarial suffixes using English-language goals paired with English targets.

Table 1 presents the ASRs when the single suffix generated in the baseline setting is reused with goals expressed in multiple languages. We aggregate results by counting a jailbreak as successful if any of the five responses—obtained by prompting the LLM with the suffix paired with goals in five different languages—constitutes a jailbreak. In this setting, the ASRs increase from 61.9%,31.9% and 38.5% to 95.6%, 85.0%, 86.5% for Mistral, Llama and Qwen models respectively. These findings demonstrate the cross-lingual reusability of adversarial suffixes. For the Mistral-7B model, the English baseline suffix demonstrates stronger transferability to Italian than to Javanese, a pattern that may reflect underlying differences in language-specific data distributions. In the case of Llama 3.1 and Qwen2.5, the baseline suffix exhibits the lowest transferability to Chinese, which may be attributable to the considerable linguistic divergence between English and Chinese, particularly in orthographic systems and typological structure.

Figures 3 and 4 present ASRs for Mistral-7B across suffix-generation strategies. With English targets (Figure 3), the *English goals* method improves ASRs by 3–5%, except for Italian, while the *ML goals* method underperforms. When the target and goal languages match (Figure 4), the *English goals* method improves ASRs only for Vietnamese and Javanese, and the *ML goals* method provides no gains. In both settings, the *Aggregation* method achieves the largest improvements (5–18%). Overall, suffixes generated with non-English goals offer language-dependent benefits, whereas aggregation consistently boosts ASRs. Similar patterns are observed for Llama and Qwen models (see Appendix).

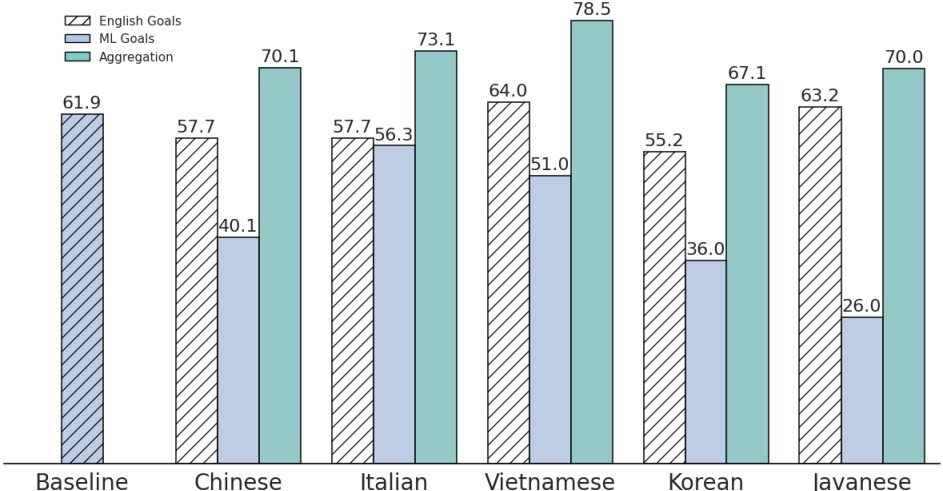

Figure 4: Multilingual GCG Jailbreaks with multilingual targets for Mistral-7B

---

**Algorithm 1** Hierarchical random sampling for language pair construction

---

1: $LangType \leftarrow$ Weighted Sampling ($\{High, Mid, Low\}$)
2: $GoalLang \leftarrow$ Uniform Sampling($\{Languagues\}$ $of$ $LangType$)
3: **if** $LangType \in \{High, Mid\}$ **then**
4:     $TargetLang \leftarrow GoalLang$
5: **else**
6:     $TargetLang \leftarrow English$                      ▷ Goals in low-res. language paired
7:                                        ▷ with English targets achieve better ASRs
8: **end if**
9: **return** $(GoalLang, TargetLang)$

---

## 3.4 Multilingual Graph Attacks

The proposed graph-based method seeks to increase attack success by automating the generation and evaluation of multilingual adversarial suffixes and exploiting their cross-lingual reusability. As illustrated in Figure 5, the graph consists of nodes corresponding to goal–target language pairs and is constructed using hierarchical random sampling. We first sample a language category from three resource tiers (high, mid, or low) and then sample a specific goal language within the selected tier. The target language is assigned as either English or the same language as the goal (self-pair), conditional on the sampled goal language type.

The proposed graph-based method starts at the root node and traverses the graph to iteratively generate and evaluate multilingual jailbreak suffixes. At each node, the GCG algorithm constructs a suffix using the goal language and target language of the node's pair, then immediately evaluates it. If unsuccessful, the traversal proceeds to the next node. The process ends when a multilingual jailbreak is found or all nodes are visited.

To further exploit suffix transferability, we propose an augmented approach, termed *graph+random*, which evaluates each generated suffix on the corresponding goal expressed in three randomly sampled languages from the language set. This increases the likelihood of attack success by reusing suffixes. The complete procedures are described in Algorithm 1 and Algorithm 2.

Figure 6 presents ASRs for three aligned LLMs using the proposed graph-based methods. The graph method increases ASRs to 91.0%, 86.0%, and 83.0% ASRs for the Mistral-7B, Llama-3.1, and Qwen2.5 models, respectively. The *graph+random* method further raises ASRs to 95.7%, 95.0%, and 89.6% for the these models.

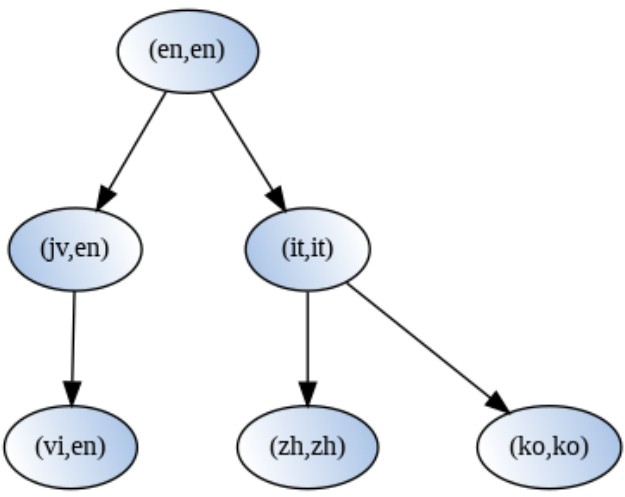

Figure 5: The graph for multilingual jailbreak

---

**Algorithm 2** Multilingual graph-based attack using GCG

---

1: $jailbreak \leftarrow False$
2: **for** each $node$ of $Graph$ **do**
3:     $\{goal\_lang, target\_lang\} \leftarrow node\_value(language\ pair)$
4:     *GCG Suffix $\leftarrow$ GCG algorithm with goal_lang and target_lang*
5:     $lang\_sets \leftarrow \{English, goal\_lang, 3\ other\ languages\ randomly\ chosen\}$
6:     **for** each $lang$ of $lang\_sets$ **do**
7:         *Generate LLM responses with the goal in $lang$+GCG Suffix*
8:         *Evaluate the responses using LLM as a judge and key word filters*
9:         **if** *response is unsafe* **then**
10:            $jailbreak \leftarrow True$
11:         **end if**
12:     **end for**
13:     *break if jailbreak*
14: **end for**

---

## 4 Multilingual AutoDAN

### 4.1 Method

We investigate two strategies for generating multilingual jailbreak prompts. ML-AutoDAN (CSW) (Algorithm 3) extends AutoDAN(Liu et al., 2024) by incorporating multilingual goals and targets while retaining the reference (prototype) prompts in English, thus employing a code-switching approach. For the ML-AutoDAN (CSW17) method, the goal and target pair language is randomly selected from one of 17 languages and the genetic algorithm refines the prompts via crossover, mutation, and selection. In contrast, the ML-AutoDAN (CSW5) method restricts the choice of languages for goals and targets to one of 5 languages that Llama 3.1 is known to support more reliably, applying the same optimization procedure.

ML-AutoDAN (*language*)(Algorithm 4) employs multilingual reference (prototype) prompts, where the genetic algorithm searches over prompts of 300–500 words written in a specified *language*. Language selection is tailored to LLM model's multilingual coverage (e.g., Llama-3.1 supports seven non-English languages, whereas Qwen2.5 supports 29 or more). In this second approach, we focus on Chinese, Italian, and Portuguese to generate multilingual jailbreak prompts. The multilingual reference (prototype) prompts are generated by translating the English reference (prototype) prompts (Mazeika et al., 2024) using an abliterated version of open LLMs, and the translations are verified via back-translation.

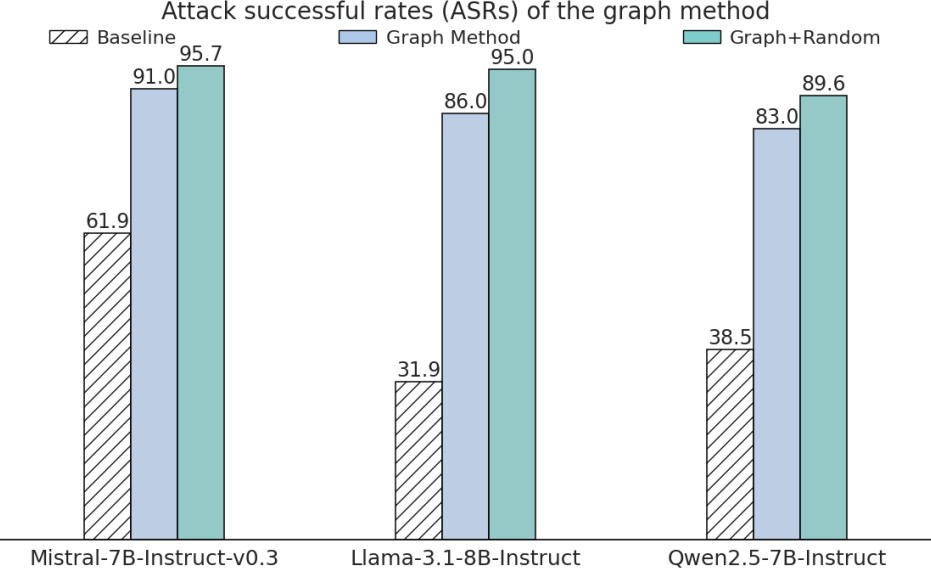

Figure 6: Graph-based multilingual Jailbreaks

---

**Algorithm 3** ML-AutoDAN (CSW)

---

1: Init reference (prototype) prompts in *English*
2: **for** *each behavior in datastes* **do**
3:   $lang \leftarrow sampling$   {5 or 17 languages}
4:   *Set the goal and target of the behavior in* $lang$
5:   *Conduct AutoDAN to find the best solution*
6:   *Generate and evaluate the LLM responses*
7: **end for**

---

### 4.1.1 Multilingual Crossover and Mutation

Genetic search over multilingual prompts requires reliable sentence segmentation and language-aware variation operators. Prior to crossover, we perform language identification and apply a language-specific sentence tokenizer to obtain valid crossover points. Mutation proceeds via two mechanisms. In *LLM-based mutation*, we prompt the model to rewrite, substitute, or otherwise perturb sentences *in the detected language* to preserve linguistic validity. In *lexical (synonym-based) mutation*, we first tokenize using language-appropriate tools (e.g., `jieba` for Chinese) and then substitute candidate tokens using language-specific lexical resources (e.g., WordNet for English). These procedures introduce diversity while maintaining fluency.

### 4.2 Multilingual AutoDAN Results

We evaluate multilingual jailbreak prompt generation on the same three open-source LLMs—Mistral-7B, Llama-3.1-8B, and Qwen2.5-7B—as used in the Multilingual GCG method.

Table 2 reports ASRs for the baseline AutoDAN and its multilingual variants across three LLMs. Mistral-7B consistently achieves the highest ASRs, with 98.8% for ML-AutoDAN (CSW17) and 99.4% for ML-AutoDAN (CSW5), outperforming LLama-3.1-8B (80.2% and 95.2%) and Qwen2.5-7B (81.7% and 81.0%). This reflects Mistral's greater vulnerability to multilingual jailbreak prompts.

LLama-3.1-8B shows marked improvement from ML-AutoDAN (CSW17) to ML-AutoDAN (CSW5), highlighting the importance of aligning language selection with the model's multilingual capabilities to enhance attack success. However, Llama-3.1-8B struggles with long Italian prompts, yielding a lower ASR of 56.2% under ML-AutoDAN (Italian).

---
**Algorithm 4** Multilingual AutoDAN (language)
---
1: $lang \leftarrow \{Chinese, Italian, Portugues\}$
2: Init reference(prototype) prompts in *lang*
3: **while** termination criteria not met **do**
4:     *Conduct multilingual crossover and mutation to generate offspring*
5:     *Evaluate the offspring*
6:     *Select individuals for the next generation*
7: **end while**
8: **return best solution found**
---

Table 2: ASR(%) for baseline and multilingual AutoDAN

| Method | Mistral-7B | Llama-3.1-8B | Qwen2.5-7B |
|---|---|---|---|
| AutoDAN (Baseline) | 97.8 | 88.5 | 65.6 |
| ML-AutoDAN (CSW17) | 98.8 | 80.2 | 81.7 |
| ML-AutoDAN (CSW5) | **99.4** | **95.2** | 81.0 |
| ML-AutoDAN (Chinese) | 97.1 | 84.0 | 74.2 |
| ML-AutoDAN (Italian) | 88.5 | 56.2 | 78.8 |
| ML-AutoDAN (Portuguese) | 98.3 | 78.2 | **93.8** |

Qwen2.5-7B attains its highest ASR (93.8%) with ML-AutoDAN (Portuguese), indicating language-specific variability in model vulnerability. Such variability likely stems from differences in the models' pretraining corpora and the data used during their respective alignment phases. For instance, Qwen2.5's high ASR with Portuguese prompts may suggest that this language was underrepresented in its safety fine-tuning data compared to others, a hypothesis that future work should further investigate.

These results highlight the influence of model-specific linguistic characteristics on Multilingual AutoDAN's efficacy and the challenge of achieving strong multilingual understanding alongside robustness.

## 5 Conclusion

In conclusion, this study provides empirical evidence that both GCG and genetic algorithm–based methods can effectively generate multilingual jailbreak prompts, albeit through different mechanisms. The Multilingual GCG approach demonstrates that short prompts can achieve high aggregated ASRs, primarily due to the cross-lingual transferability of adversarial suffixes. Conversely, the Multilingual AutoDAN framework highlights the capacity of genetic algorithms to automatically produce long multilingual prompts with high ASRs. Taken together, these results not only advance the current understanding of multilingual jailbreak vulnerabilities but also expose critical gaps in the robustness of large language models (LLMs) across linguistic boundaries. By illuminating these vulnerabilities, our findings underscore the urgency of developing comprehensive multilingual safety frameworks and resilient defense mechanisms, thereby contributing to the broader agenda of ensuring the trustworthy and secure deployment of LLMs in global contexts.

## 6 Limitation and Future Work

Our study automates the generation of multilingual jailbreak prompts for a *single* LLM and does not evaluate the transferability of these prompts across different models; assessing cross-model robustness is an important direction for future work. In addition, we focus on eliciting harmful text responses from text-only LLMs; extending automated multilingual jailbreak prompt generation to settings that involve multimodal LLMs represents another promising avenue.

## Ethics Statement

We place strong emphasis on the ethical dimensions of our work. This study centers on improving the safety of large language models—specifically addressing multilingual jailbreak attacks—via automatic multilingual prompt generation. Our approach aims to substantially reduce unsafe responses produced by LLMs.

All experiments utilize publicly available, open datasets. Results and conclusions are reported with rigor and objectivity, adhering to best practices for scientific integrity. Consequently, we believe the research poses no ethical concerns.

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

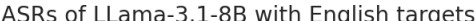

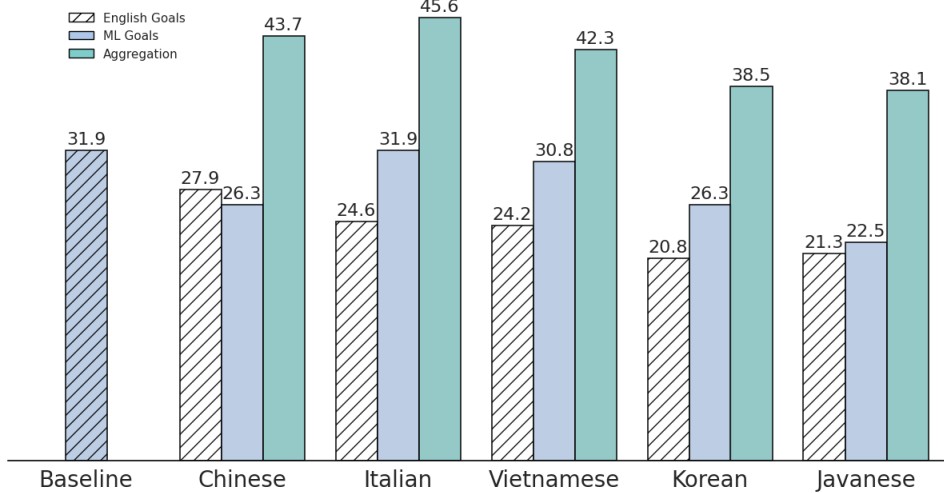

Figure 7: Multilingual GCG Jailbreaks with English targets for Llama 3.1-8B

## A Technical Appendices and Supplementary Material

### A.1 More Experiments on Multilingual GCG

Figures 7 and 8 report ASRs for the Llama-3.1 model across suffix-generation strategies. With English targets (Figure 7), the *English goals* and *ML goals* methods yield negligible gains. When the target and goal languages match (Figure 8), the *English goals* method improves ASRs by 1–6% for most languages except Javanese, while the *ML goals* method yields 6–7% gains for Italian and Vietnamese but none for others. In both settings, the *Aggregation* method produces the largest improvements, boosting ASRs by 6–25%. Overall, non-English goals offer language-dependent benefits, whereas aggregation consistently enhances performance.

Figures 9 and 10 report ASRs for the Qwen-2.5 model across suffix-generation strategies. With English targets (Figure 9), the *English goals* method yields modest gains of 0.7–1.5% for Chinese, Vietnamese, and Korean, while the *ML goals* method generally underperforms. When the target and goal languages match (Figure 10), neither method provides measurable improvements. In both settings, the *Aggregation* method delivers the largest gains, boosting ASRs by 1–13% for most languages. Overall, non-English goals offer limited benefits for Qwen-2.5, whereas aggregation consistently enhances performance, except when both the target and goal are Javanese.

We further examined the effect of decoding methods by aggregating two runs of the baseline. With `do_sample` set to `False`, aggregation produced less than a 0.2% difference in ASR. In contrast, with `do_sample` set to `True` and *temperature* fixed at 0.7, aggregation increased ASRs by up to 5%. This effect is likely due to suffixes that lie near the decision boundary of successfully jailbreaking the model.

### A.2 Experiment Setup

The experiments were conducted on a GPU cloud instance equipped with a single NVIDIA RTX 4090 (24 GB) GPU. The software stack included Python 3.11 or later, PyTorch 2.6.0 (necessary for loading the `model.bin` file) with CUDA 12.4, and the HuggingFace Transformers library for model loading and inference. All computations were performed in half-precision (torch.float16) to optimize memory usage and computational efficiency. For faster execution, more powerful GPUs such as the NVIDIA A100 can be utilized.

Random seeds were fixed for all experiments. The complete codebase, including prompt generation and evaluation scripts, will be released to facilitate reproducibility. For decoding, we used a temperature of 1.0 for Mistral, and a temperature of 1.0 with a repetition penalty of 1.5 for the LLama

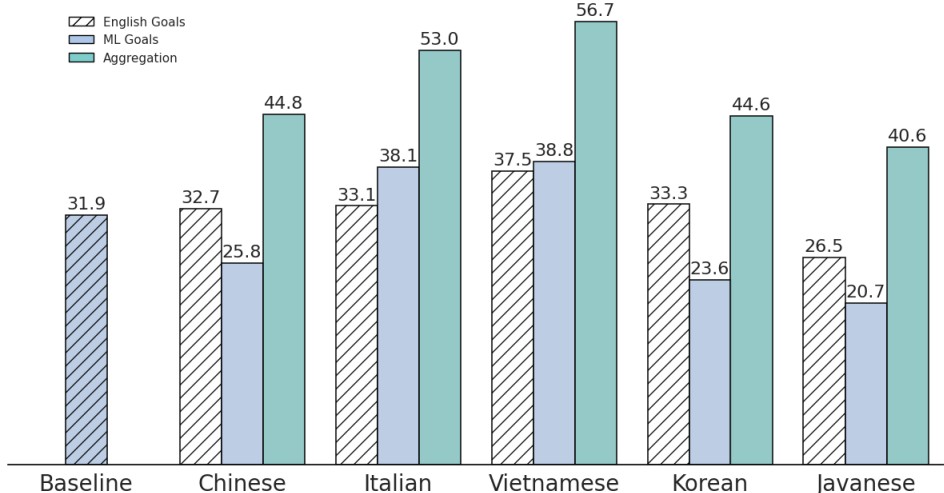

Figure 8: Multilingual GCG Jailbreaks with multilingual targets for Llama 3.1-8B

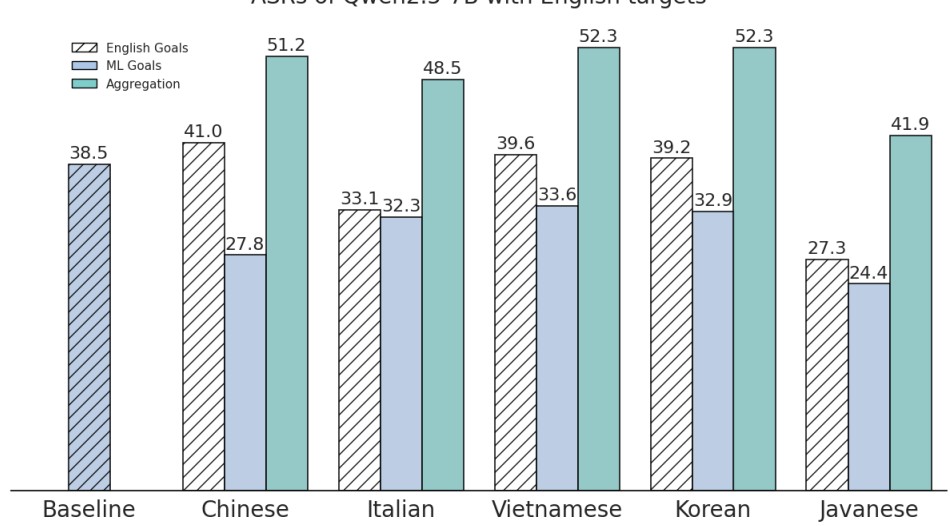

Figure 9: Multilingual GCG Jailbreaks with English targets for Qwen2.5-7B

and Qwen models. The `do_sample` parameter was set to `False`. For the 520 harmful behaviors considered, the multilingual GCG graph attack required approximately 10–12 hours per run, with runtime variation attributable to both the stochasticity of the graph construction, differences in model robustness to adversarial suffix attacks, and the number of graph nodes.

Multilingual AutoDAN (synonym-based mutation) required 3–10 GPU-hours with early stopping and runtime evaluation enabled, depending on model vulnerability. Runtime was substantially influenced by the choice of prototype prompts: stronger prototypes required fewer iterations to jailbreak the model, leading to faster completion. The same decoding settings (*temperature* = 1, `do_sample` = False) were applied to Multilingual AutoDAN.

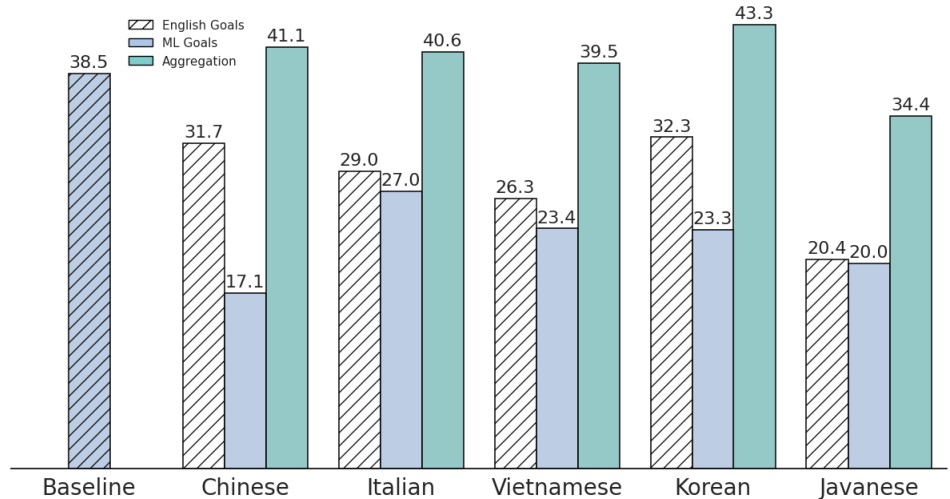

Figure 10: Multilingual GCG Jailbreaks with multilingual targets for Qwen2.5-7B

