# OpenReview forum: "Automated Generation of Multilingual Jailbreak Prompts"
_NeurIPS.cc/2025/Workshop/Reliable_ML — NeurIPS 2025 - Reliable ML Workshop_

### Official Review · Reviewer_qTLB · 2025-09-12
**Interesting analysis of multilingual jailbreaking techniques**

**Rating:** 7
**Confidence:** 4

**Review:**

1. Summary:
The authors explore two methods of automating the generation of multilingual (non-English) jailbreaking prompts. They red-team models by developing a graph-based attacking technique to jailbreak the model through different languages. They show strong empirical evidence of eliciting harmful outputs from open-source LLMs through the GCG and genetic algorithm-based methods.

2. Strengths:
(A) Strong connection to previous work (ie. discussion and analysis of multilingual AutoDAN and multilingual GCG). The novel algorithmic implementation clearly builds off of existing methods.
(B) It was helpful that you established performance success on baselines before conducting your analyses.
(C) There is strong empirical evidence that ASRs increase across models when multiple languages are considered.
(D) The graph traversal method for generating adversarial prompts is a novel contribution.
(E) ASR values are extremely high (in the 90s or high 80s) in the new jailbreaking method.

3. Weaknesses:
(A) Evaluation design may inflate ASR. The Aggregation metric counts success if any of multiple language queries fires; this can overstate per-query risk. Per-language and per-attempt ASRs should be primary, with aggregation as a secondary metric.
(B) Crucial details are missing/under-specified: the exact 17 languages, translation quality controls, judge model(s)/prompts, seeds, and code. Without these, replication is hard and results could depend on hidden choices.

4. Suggestions:
(A) Minor suggestion: The abstract is quite long! It would be useful to condense it and focus on takeaways rather than specifications.
(B) When you introduce the dataset, please include the dataset size. What are the labels? How are the items in the dataset distributed by language (ie. uniformly)?
(C) How exactly is "Aggregation" (Table 1) calculated? Include a table caption, as well.
(D) What is meant by "LangType" and "GoalLang"? It would be useful to define those before Figure 4 appears. It is unclear what is meant by resource tiers.
(E) It would be useful for you to release a dataset of your new collection of adversarial prompts (generated via. your novel algorithm) publicly.
(F) It would be helpful for you to include a figure that shows one traversal through the graph and the prompt that is built (choose the languages that are traversed for concreteness).

5. Ethics
(A) Given the high effectiveness of your proposed methods, I would like to see an analysis of mitigation steps or a method to control against your new method.

---

### Official Review · Reviewer_BcCE · 2025-09-20

**Rating:** 8
**Confidence:** 4

**Review:**

The work considers the problem of automatically generating multilingual jailbreak prompts. The goal behind this line of research involves identifying vulnerabilities of aligned LLMs not just in english, but in other languages as well, where the topic of making models robust to jailbreak prompts has not been explored as much. The paper proposes multilingual variants of some well-known methods for jailbreak prompt generation (namely GCG and AutoDAN), as well as novel graph-based whose purpose is to automate the attack. Specifically, the graph-based method involves the adversary traversing a graph consisting of nodes with language pairs, and using that to generate multilingual jailbreak prompts.

I consider this to be a good paper. It focuses on a topic that is of great interest (aligning LLMs with human values is an important research direction in my opinion), and advances known methods by introducing their multilingual analogues. I found the experimental evaluation to be comprehensive, which is also a strong point of the paper. For all the above reasons, I recommend acceptance.

---

### Official Review · Reviewer_gowB · 2025-09-20
**An effective set of algorithms that might be worth presenting**

**Rating:** 5
**Confidence:** 2

**Review:**

### Summary:
This paper generalizes work on autonomously generating jailbreak prompts to generating jailbreaks in various languages. They introduce a graph based algorithm for generating attacks. They test these generation techniques on a range of models.

### Strengths:
There seems to be a large improvement in the success rates of AutoDAN and GCG when allowing for various languages to be used, which came as a surprise to me. The rate of improvement alone could make this paper worth presenting at this workshop.

### Weaknesses:
The algorithms introduced in this work seem to me to be, at their core, wrappers of existing techniques with translation.

Moreover, the graph algorithm the authors introduce is very strange, there doesn't seem to be a need for a graph at all. The algorithm randomly samples language pairs to search over, then runs GCG on translations. As far as I can tell, this never uses or needs the structure of nodes and edges anywhere. That being said, this work is very far from my area of expertise, so it is likely I've missed something.

### Suggestions for authors:
Figures 1 and 2 are confusing to me, it would be helpful if a caption was used. And Figure 5 is too low a resolution.